# Indigenous Nutrient Supplying Capacity of Young Alluvial Calcareous Soils Favours the Sustainable Productivity of Hybrid Rice and Maize Crops

Shiveshwar Pratap Singh [1,*], Sudarshan Dutta [2,*], Shankar Jha [1], Shiv Shankar Prasad [1], Sanjay Kumar Chaudhary [3], Madhab Chandra Manna [1], Kaushik Majumdar [4], Prashant Srivastava [5], Pothula Srinivasa Brahmanand [6], Krishna Murari Singh [7] and Krishna Kumar [8,*]

[1]  Department of Soil Science, Dr. Rajendra Prasad Central Agricultural University, Pusa, Samastipur 848125, Bihar, India
[2]  Kosher Climate India (P) Ltd., Bangaluru 560102, Karnataka, India
[3]  Department of Agronomy, Dr. Rajendra Prasad Central Agricultural University, Pusa, Samastipur 848125, Bihar, India
[4]  African Plant Nutrition Institute, Lot 660, Hay Moulay Rachid, Ben Guerir 43150, Morocco
[5]  Commonwealth Scientific and Industrial Research Organization (CSIRO) Land and Water, Urrbrae, SA 5064, Australia
[6]  Directorate of Research, Dr. Rajendra Prasad Central Agricultural University, Pusa, Samastipur 848125, Bihar, India
[7]  Post-Graduate College of Agriculture, Dr. Rajendra Prasad Central Agricultural University, Pusa, Samastipur 848125, Bihar, India
[8]  Pt. Deen Dayal Upadhyay College of Horticulture and Forestry, Dr. Rajendra Prasad Central Agricultural University, Pusa, Samastipur 848125, Bihar, India
*  Correspondence: sp26814@gmail.com (S.P.S.); sudarshand@kosherclimate.com (S.D.); kkpath@gmail.com (K.K.)

**Abstract:** The crop productivity in calcareous soils is low due to their low organic matter content, high pH levels and improper nutrient management without considering the indigenous nutrient supplying capacity and crop yield potential; therefore, this study was conducted for a quantitative assessment of the nutrient supplying capacity of a calcareous soil on the productivity of hybrid and conventional rice and maize crops using an omission plot technique. The treatments included the ample application of Nitrogen (N), Phosphorus (P), Potassium (K), Sulphur (S) and Zinc (Zn), and an unfertilized check and omissions of N, P, K, S and Zn in rice and maize for six cropping seasons. The impact of the nutrient omission towards crop productivity was highest for nitrogen followed by phosphorous, potassium, zinc and sulphur. The total grain yield (3 yr average) in the hybrid rice–maize system was highest (16.32 t ha$^{-1}$) for the optimum fertilized plot and lowest (6.34 t ha$^{-1}$) for the unfertilized check. The sustainable yield index indicated that hybrid and conventional rice-maize cropping systems were more sustainable in the amply fertilized plot than in the nutrient-limited and unfertilized treatment plots. The average percent contributions of nitrogen, phosphorous, and potassium from the soil towards total nutrient removal were 36, 80 and 137 kg ha$^{-1}$, in the hybrid system and 24, 54 and 104 kg ha$^{-1}$ in the conventional system, respectively. The return on investment (ROI) of the N, P, K, S and Zn for the hybrid rice was 21.2, 7.1, 6.7, 4.1, and 0.3 USD, respectively, while for the maize it was 28.8, 7.6, 4.9, 6.5, and 0.7 USD, respectively. The results suggest that there is a direct link between the soil nutrient supplying capacity and the nutrient requirements by different types of crops in calcareous soil; therefore, the omission plot technique used for the assessment of the indigenous nutrient supplying capacity could be used in the larger domain for improved nutrient management, through synchronization with a targeted crop yield for improved productivity, soil fertility, nutrient use efficiency and farm income.

**Keywords:** calcareous soils; omission plot technique; return on investment; nutrient supplying capacity

## 1. Introduction

Rice–maize rotation is one of the major cropping systems in South Asia [1]. Rice (*Oryza sativa* L.) is an important staple food for more than 750 million of the world's poorest people [2]. On the other hand, the area used for maize (*Zea mays* L.) is increasing and it is widely cultivated over 150 million ha across 160 countries. With its highest genetic yield potential, maize contributes to 36% (78.2 million tonnes) of the total grain production of the world, and is referred to as the 'queen of cereals' [3–6]. In India, maize is predominantly utilized by industry with around 25% of its production being consumed as human food [7].

The main constraints in the achievement of an attainable yield in a rice–maize system are the insufficient application of plant nutrients, improper farming practices, a limited reuse of farm wastes, and intensive crop cultivation. The decline in the productivity efficiency of applied nutrients is one of the important reasons for stagnation in the productivity of the crops [8,9]. Improvements in productivity from the existing conditions are warranted for global food and nutritional security; however, this is a challenging task due to the limited available options [10].

In the state of Bihar, the average productivity of *kharif* rice (2.18 t h$^{-1}$) and *winter* maize (3.38 t ha$^{-1}$) is lower than the national productivity (*kharif* rice 2.62 t ha$^{-1}$ and *winter* maize 4.63 t ha$^{-1}$); although, the yield potential of these crops is very high. Hybrid rice has yield advantages of more than thirty percent over conventional varieties while hybrid maize can provide a >60% yield over the existing yields with conventional varieties [11,12]. Improved nutrient management is one of the most effective approaches to increase cereal production [12,13], that could ensure high yields along with good quality produce [14]. In general, insufficient or imbalanced fertilizer nutrients being applied to soil are not based on the soil's nutrient-supplying capacity. Farmers apply excess nitrogenous fertilizers due to its quick visual impact on crops and its lower price than the other fertilizer nutrients, which causes an increase in the deficiency of other nutrients, such as P and K [15]. The inherent capacity of soil to supply N, P, and K and subsequent fertilizer application requirements vary among farmers' fields because of variable, long-term management practices. The nutrient availability in farmers' fields can be assessed using the "Omission Plot" technique, where a particular essential nutrient is omitted from the fertilization schedule while keeping the supply of other limiting nutrients at an optimum quantity [16].

In India, the estimated area of calcareous soils is 228.8 m ha, which covers 69.4% of the total geographical area of the country [17]. Calcareous soils, classified as calciorthents [18], are those that contain sufficient calcium or magnesium carbonate to effervesce visibly when treated with cold 0.1 *N* hydrochloric acid [19]. The pH of calcareous soils varies from 7.0 to 8.4 because of the limited solubility of $CaCO_3$ and plant growth is generally restricted by deficiencies of essential nutrients, particularly the micronutrients [12,19,20]. Increasing nutrient deficiencies due to their over-mining is another major problem [21]. Within the intensively cultivated calcareous soils, widespread multi-nutrient deficiencies including N, P, K, S, Zn and Band sporadic deficiencies of iron, manganese, copper, etc., are the important constraints for improvements in a crop yield [22].

The natural productivity of the Indo-Gangetic Plain (IGP) soils is low; however, the productivity potential for the rice–maize cropping system is very good, making the diagnosis of nutrient requirements and the supply of nutrients to crops in economically optimum amounts even more critical. Now farmers are shifting towards hybrid crops for obtaining more yield and income. Hybrid crops are heavy feeders and, thus, will cause more mining of nutrients from the soil; therefore, nutrient management based on the soil supplying capacity will help in minimizing a deterioration in soil health. Considering this, the present study was conducted to assess the nutrient response (N, P, K, S and Zn) on hybrid rice (wet season) and hybrid maize (winter season) in nutrient omission trials, to compare the yield variations in hybrid and conventional crops and to evaluate the nutrient supplying capacity of soil.

## 2. Materials and Methods

### 2.1. Experimental Site and Climate

The present trial was conducted at the research farm of Dr. R. P. Central Agricultural University, India. The experimental site was located at the southern bank of the *Burhi Gandak* river (N 25°58′43.3″ and E 085°40′24.5″ and 52.3 m above the mean sea level). At this location, the climate is sub-humid sub-tropical. It falls in the south-west region and the monsoon, generally, starts from the middle of June and continues up to the first week of October. The annual average rainfall is about 1200 mm, and more than 80% is received during June and October. The weather data (e.g., rainfall, the minimum and maximum temperatures, and relative humidity) for the crop seasons (*Kharif*-2013 to *Rabi*-2015–16) are presented in Figure 1. The soil status of the experimental field is depicted in Table 1.

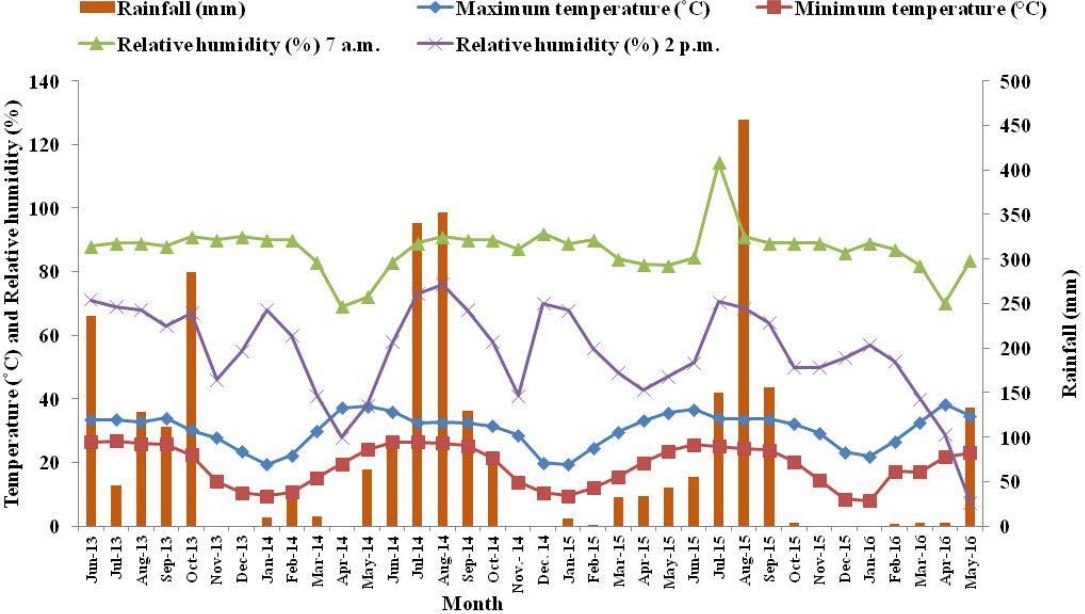

**Figure 1.** Monthly weather data for the study period.

**Table 1.** Initial status of experimental soil (0–15 cm).

| Variables | Status | Reference/Method |
|---|---|---|
| Sand (%) | 25.80 | |
| Silt (%) | 52.53 | International Pipette Method [23] |
| Clay (%) | 20.58 | |
| Textural class | Silty loam | |
| Bulk density (Mg m$^{-3}$) | 1.26 | Core Method [24] |
| pH (1:2, soil:water) | 8.28 | Glass Electrode pH meter [25] |
| EC (dSm$^{-1}$) at 25 °C | 0.47 | Conductivity Bridge [25] |
| Organic carbon (%) | 0.33 | [26] |
| CaCO$_3$ (%) | 21.5 | [27] |
| Potassium permanganate extractable N (kg ha$^{-1}$) | 212.80 | Alkaline Permanganate Method [28] |
| Sodium bicarbonate extractable P (kg ha$^{-1}$) | 16.99 | Olsen Method [29] |
| Ammonium acetate extractable K (kg ha$^{-1}$) | 86.24 | Flame Photometer [25] |
| Calcium chloride extractable S (mg kg$^{-1}$) | 52.81 | 0.15% Calcium Chloride Method [30] |
| DTPA extractable Zn (mg kg$^{-1}$) | 0.72 | [31] |

The nutrient omission experiment was conducted in a randomized block design with three replications. The net plot size was 18 m$^2$ (6 m × 3 m) for each treatment and each replication. During *Kharif* (wet) seasons (2013, 2014 and 2015), two varieties of rice, viz., hybrid rice (cv. Arize 6444) and conventional rice (cv. Rajshree) were grown, whereas during *Rabi* (dry) seasons (2013–14, 2014–15 and 2015–16), two varieties of maize, viz.,

hybrid maize (cv. DKC-9081) and conventional maize (Laxmi) were grown (Table 2). The hybrid and conventional rice at 20 and 35 kg ha$^{-1}$, respectively, were sown on seedbeds, and 20 days-old-seedlings at 2–3 seedlings per hill were transplanted manually with a row-to-row distance of 20 cm and plant-to-plant distance of 10 cm. The hybrid and conventional maize was seeded manually with a seed rate of 20 kg ha$^{-1}$ and a plant-to-plant distance of 20 cm and row-to-row distance of 60 cm. Fertilizer treatments were applied in nine different combinations (Table 2). The optimum fertilizer rates for the conventional and hybrid rice were 125-50-60-30-3 and 175-70-80-30-3 kg of N, $P_2O_5$, $K_2O$, S, and Zn ha$^{-1}$, based on yield targets of 5 and 7 t ha$^{-1}$, respectively (Table 3). For the maize, the ample rates were 150-70-120-30-3 and 210-140-200-30-3 for conventional and hybrid yield targets of 6 and 10 t ha$^{-1}$, respectively (Table 3). The fertilizer sources used were urea (46% N), triple superphosphate (46% $P_2O_5$), muriate of potash (60% $K_2O$), bentonite-S (90% S), and Zn-EDTA (12% Zn). The N rate for both crops was split into three applications, whereas the K rate was split into two applications (Table 3). Before the rice transplanting, and the sowing of maize, the field was prepared manually with the spade and all other nutrients were applied and mixed in the soil just prior to planting, similar to conventional tillage practices (Table 3). The impact of the nutrient omission was studied only in the hybrid crops (Table 2), whereas the conventional crop was grown to compare the yield gap and nutrient uptake differences between the two varieties.

**Table 2.** Treatments details.

| Treatments | Crop/Variety | | Code |
|---|---|---|---|
| | *Kharif* **Rice** | *Rabi* **Maize** | |
| N + P + K + S + Zn | Hybrid (Arize-6444) | Hybrid (DKC-9081) | $T_1$ |
| Hybrid rice under unfertilized check | Hybrid (Arize-6444) | Hybrid (DKC-9081) | $T_2$ |
| P + K + S + Zn (-N) | Hybrid (Arize-6444) | Hybrid (DKC-9081) | $T_3$ |
| N + K + S + Zn (-P) | Hybrid (Arize-6444) | Hybrid (DKC-9081) | $T_4$ |
| N + P + S + Zn (-K) | Hybrid (Arize-6444) | Hybrid (DKC-9081) | $T_5$ |
| N + P + K + Zn (-S) | Hybrid (Arize-6444) | Hybrid (DKC-9081) | $T_6$ |
| N + P + K + S (-Zn). | Hybrid (Arize-6444) | Hybrid (DKC-9081) | $T_7$ |
| Conventional variety under unfertilized check | Conventional (Rajshree) | Conventional (Laxmi) | $T_8$ |
| Conventional variety under ample fertilization | Conventional (Rajshree) | Conventional (Laxmi) | $T_9$ |

**Table 3.** Fertilizer dose and time of application.

| Yield Target (t/ha) | N (kg/ha): 3 Splits | | | | Total $P_2O_5$ (kg/ha) Basal | $K_2O$ (kg/ha): 2 Splits | | | Total S (Basal) | Total Zn (Basal) |
|---|---|---|---|---|---|---|---|---|---|---|
| | Total N | Basal N | AT | PI | | Total $K_2O$ | Basal | PI | | |
| | | | | | | | Rice | | | |
| 5 (conventional rice) | 125 | 55 | 35 | 35 | 50 | 60 | 30 | 30 | 30 | 3 |
| 7 (hybrid rice) | 175 | 75 | 50 | 50 | 70 | 80 | 40 | 40 | 30 | 3 |
| | | | | | | | Maize | | | |
| Yield target (t/ha) | Total N | Basal N | At V6 | At V10 | Total $P_2O_5$ (kg/ha) basal | Total $K_2O$ | Basal | At V10 | Total S (basal) | Total Zn (basal) |
| 6 (conventional maize) | 150 | 50 | 50 | 50 | 70 | 120 | 60 | 60 | 30 | 3 |
| 10 (hybrid maize) | 210 | 70 | 70 | 70 | 140 | 200 | 100 | 100 | 30 | 3 |

V6 and V10: six and ten leaf stage in maize, respectively. AT: active tillering stage. PI: panicle initiation stage.

### 2.2. Sample Collection, Preparation, and Analysis

The rice and maize were harvested manually from the entire plot. After sun drying for 3–4 days, the total biomass was recorded with an electronic balance and threshed manually to separate the grains and the straw. After recording the yields, from each treatment plot about 100 g of grain and straw samples were drawn. These samples were washed in an acidified solution and then with de-ionized water. The washed plant samples were dried in air. The straw samples were chopped into small pieces using stainless steel scissors. The plant samples were then dried in a forced-air circulation oven at $60 \pm 5$ °C.

The dried samples were pulverized in a wiring blender, which was cleaned with a hairbrush after grinding each sample. The total nitrogen in the grounded samples was determined by a modified Kjeldahl's method as described by Bremner and Mulvaney [32]. For the determination of other nutrient concentrations, 0.5 g of grounded plant samples were digested in a di-acid (9:4 *v/v*) of nitric acid ($HNO_3$)/perchloric acid ($HClO_4$). Following the digestion, the sample volumes were brought up to 50 mL using Ultra-pure MilliQ water. The total phosphorus concentration in the grain and straw was determined by the vanado-molybdo-phosphate method [33] and the absorbance was recorded with a spectrophotometer. The potassium concentration was estimated with the help of a flame photometer. The sulphur concentration was determined by the turbidimetric method, the absorbance was recorded with the spectrophotometer, and the zinc by an atomic absorption spectrophotometer as described by Tandon [34].

Soil samples (0–15 cm) were collected after the final maize harvest (2016) from each plot using a stainless-steel tube auger. The air-dried soils were grounded and sieved with a 2 mm stainless steel sieve and analysed by using standard methods (Table 1).

*2.3. Observations and Calculations*

The impact of the nutrient omission on the nutrient response (NR), agronomic efficiency (AE), apparent recovery efficiency (AR), physiological efficiency (PE), reciprocal internal use efficiency (RIUE), economic loss, and return on investment (ROI) were calculated using the following equations [35–38]:

$$\text{Nutrient Response } \left(\text{kg ha}^{-1}\right)$$
$$= \text{Grain yield in fertilized plot } \left(\text{kg ha}^{-1}\right) - \text{Grain yield in nutrient omitted plot } \left(\text{kg ha}^{-1}\right)$$

$$\text{Agronomic Efficiency } \left(\text{kg kg}^{-1} \text{ nutrient}\right) = \frac{\text{Nutrient Response (kg)}}{\text{Nutrient applied (kg)}}$$

$$\text{Apparent Recovery Efficiency (\%)} = \frac{\text{Nuf} - \text{Nuo}}{\text{Na}} \times 100$$

where Nuf is the total nutrient uptake by a crop in a fertilized plot (kg); Nuo is the total nutrient uptake by a crop in the nutrient omission plot (kg), Na is the nutrient applied (kg).

$$\text{Physiological Efficiency } = \frac{\text{NR}}{(\text{Nuf} - \text{Nto})}$$

where NR is the nutrient response; Nuf is the total nutrient uptake by a crop in a fertilized plot (kg); Nuo is the total nutrient uptake by a crop in the nutrient-omitted plot (kg).

$$\text{Reciprocal Internal Use Efficiency } = \frac{\text{Nu}}{\text{Y}}$$

where, Nu is the total absorbed nutrients by the crop (kg) and Y is the grain yield (tons).

$$\text{Return on Investment } = \frac{(\text{NR} \times \text{MSP})}{(\text{Na} \times \text{Nc})}$$

where, NR is the nutrient response (kg ha$^{-1}$), MSP is the minimum support price of grain (USD kg$^{-1}$), Na is the total nutrient applied (kg ha$^{-1}$) and Nc is the nutrient cost (USD kg$^{-1}$). The minimum support price for rice grain during 2013, 2014 and 2015 was 0.24, 0.23 and 0.23 USD kg$^{-1}$, respectively, while for maize grain it was 0.22, 0.21 and 0.20 USD kg$^{-1}$, respectively. The cost of the nutrients, nitrogen, phosphate and potash was 0.21, 0.55 and 0.47 USD kg$^{-1}$, respectively.

### 2.3.1. Sustainable Yield Index

On the basis of the yield of the rice–maize system for all the fertilizer treatments during each year, a sustainable yield index (SYI) was calculated by using the following equation [39,40]:

$$\text{SYI} = \frac{\text{Yav} - \sigma}{\text{Ymax}}$$

where Yav is the average treatment yield; $\sigma$ is the standard deviation in the treatment; Ymax is the maximum yield in the experimental trial.

The data observed from the experiment were further used to calculate the nutrient contribution from the soil as per the procedure explained by Ramamoorthy and Velayutham [41] and Prasad et al. [42]:

$$\text{CS(\%)} = \frac{\text{Nuc}}{\text{STV}} \times 100$$

where CS is the percent contribution of $N/P_2O_5/K_2O$ from soil; Nuc is the total $N/P_2O_5/K_2O$ accumulation (kg ha$^{-1}$) in the control plot; STV is the available $N/P_2O_5/K_2O$ (kg ha$^{-1}$) in the soil under the control plot.

### 2.3.2. Rice Equivalent Yield (REY)

The maize yield was converted into a rice equivalent yield (REY) and the system yield (t ha$^{-1}$) was calculated using the following formula:

$$\text{REY} \left( \text{t ha}^{-1} \right) = \frac{\text{Maize yield} \left( \text{t ha}^{-1} \right) \times \text{MSP of Maize} \left( \text{US\$ t}^{-1} \right)}{\text{MSP of Rice} \left( \text{US\$ t}^{-1} \right)}$$

$$\text{System yield} \left( \text{t ha}^{-1} \right) = \text{Rice yield} \left( \text{t ha}^{-1} \right) + \text{REY of Maize} \left( \text{t ha}^{-1} \right)$$

where MSP is the minimum support price (as given in Section 2.3).

### 2.3.3. Statistical Analysis

The experimental data were analysed with a free online agricultural data analysis tool, OPSTAT [43]. The mean values of the treatments were separated by a Fisher's protected least significant difference (LSD) test at $p \leq 0.05$.

## 3. Results

### 3.1. Impact of Nutrient Management on the Crop Yield and System Productivity

Nutrient management had substantial effects on the grain and biomass yields of the rice and maize. The omission of each nutrient, namely, nitrogen, phosphorous, potassium, sulphur and zinc, resulted in a decreased yield, with nitrogen being the most limiting nutrient for both crops ($p \leq 0.05$; Table 4).

The three-year average grain and biomass yields of the hybrid rice varied from 2.97 and 7.14 t ha$^{-1}$ under the unfertilized check (control) to 6.81 and 13.83 t ha$^{-1}$ in the fully fertilized (175-70-80-30-3 kg N, $P_2O_5$, $K_2O$, S, and Zn ha$^{-1}$) plot, respectively (Table 4). The nutrient response (e.g., a decrease in the grain yield compared with the fertilized plot) was the highest for the unfertilized control (56.4%), followed by a nitrogen, phosphorous, potassium, sulphur, and zinc (8.7%) omission (Table 4). In the hybrid rice, the harvest index varied from 0.41 (unfertilized control) to 0.49 (optimum fertilized), with significant decreases ($p \leq 0.05$) being observed in the N-omitted and unfertilized control plots compared with the optimum fertilized plot (Table 4). The hybrid rice yielded higher than the conventional rice with optimum fertilization. The nutrient response was higher for the hybrid rice (56.4%) than for the conventional rice (54%) in the unfertilized control plot over the optimum fertilized plot (Table 4). The highest reduction over the 3 yr period in the

hybrid rice grain yields was due to a nitrogen omission (49%) followed by phosphorous (17%), potassium (16%), and sulphur and zinc (both 9%).

The grain yield of the hybrid maize ranged from 3.76 t ha$^{-1}$ in the unfertilized check plot to 9.66 t ha$^{-1}$ in the amply fertilized plot (Table 4). The highest nutrient response was observed in the unfertilized control plot (61.1%) followed by the nitrogen, phosphorous, potassium, zinc and sulphur omitted plots (Table 4). The harvest index varied from 0.45 (unfertilized check) to 0.51 t ha$^{-1}$ (optimum fertilized). The grain yield of the hybrid maize diminished by 55.6% in the N omission plot followed by decreases due to the omission of P (24.8%), K (19.7%), Zn (12.3%) and S (8.4%) compared with the amply fertilized plot.

The total grain yield (3 yr average) in the hybrid rice–maize system was highest (16.32 t ha$^{-1}$) for the optimum fertilized plot and lowest (6.34 t ha$^{-1}$) for the unfertilized check plot (Table 4). The nutrient response was highest in the unfertilized plot (61.1%) followed by the nitrogen, phosphorous, potassium, zinc and sulphur omitted plots (Table 4). The system yield of the conventional variety was lower than the hybrid variety. The sustainable yield index (SYI) for the system was highest for both the hybrid (0.89) and conventional crops (0.88) in the amply fertilized plots and lowest in the unfertilized plots (Table 4). In the nutrient omission plots, the SYI of the system was lowest for N (0.33) followed by P (0.61), K (0.66), Zn (0.74) and S (0.79) (Table 4).

### 3.2. Nutrient Uptake and Content of Rice and Maize

The omission of nitrogen, phosphorous, potassium, sulphur and zinc significantly decreased ($p < 0.05$) the respective nutrient content in both the hybrid rice and maize grains and the biomass, compared to that with the amply fertilized plot (Table 5a,b). The contents of nitrogen, phosphorous, potassium, sulphur and zinc in the amply fertilized conventional crops were lower than those of the hybrid crops. The 3 yr average contents of nitrogen, phosphorous, potassium, sulphur (%) and zinc (mg kg$^{-1}$) in the hybrid rice grain ranged from 1.03 to 1.45, 0.179 to 0.234, 0.165 to 0.225, 0.092 to 0.110 and 28.7 to 33.9 in the amply fertilized and check plots, respectively (Table 5a,b). The nitrogen, phosphorous, potassium, sulphur (%) and zinc (mg kg$^{-1}$) in the hybrid maize grain ranged from 0.67 to 1.06, 0.106 to 0.193, 0.441 to 0.705, 0.082 to 0.132 and 25.6 to 43.8, respectively (Table 5a,b).

In general, for the 3 yr average nutrient accumulation by the hybrid maize and rice crops, a significant decrease was observed in the nutrient-omitted and unfertilized control plots (Table 6). The decline in the nutrient accumulation following the nutrient omission was more prominent in the hybrid maize than the hybrid rice crop. The nitrogen, phosphorous, potassium, sulphur and zinc accumulation by the hybrid rice varied from 42.6 to 126.8, 7.6 to 21.1, 63.8 to 139.1, 7.0 to 15.8 kg ha$^{-1}$ and 214.9 to 496.2 g ha$^{-1}$, respectively, whereas in the hybrid maize, this ranged from 33.0 to 156.5, 6.1 to 31.5, 54.5 to 214.4, 4.7 to 22.10 kg ha$^{-1}$ and 174.2 to 943.8 g ha$^{-1}$, respectively (Table 6). In the nutrient-omitted plots, the nitrogen, phosphorous, potassium, sulphur and zinc uptake by the hybrid rice was found to be decreased by 57.0, 32.2, 31.0, 18.6 and 18.1%, respectively, compared with the amply fertilized plot, whereas the uptake of these nutrients decreased by 70.0, 52.3, 44.5, 36.0, and 47.3% due to the nutrient-omission in the hybrid maize. The decrease in the total nutrient uptake by the system was greatest in the N-omitted plot (64.2%) followed by P (44.3%), K (39.2%), S (28.8%) and Zn (37.2%).

### 3.3. Nutrient Use Efficiencies

The use efficiencies for each nutrient were calculated for the crops using the amply fertilized plot and respective nutrient-omitted plots. The agronomic efficiency (AE), physiological use efficiency (PE) and apparent recovery (AR) were generally higher in the second and third years than in the first year in both the rice and maize (Table 7). The 3 yr average AE (kg kg$^{-1}$) of the nitrogen, phosphorous, potassium, sulphur and zinc of the hybrid rice was 19.1, 16.7, 13.6, 19.7 and 197.6, respectively (data not depicted in table). The average agronomic efficiency of the nitrogen, phosphorous, potassium, sulphur and

Zn in the hybrid maize was higher than the hybrid rice and was 27.1, 18.7, 10.4, 33.0 and 449.6 kg kg$^{-1}$, respectively (data not depicted in table).

**Table 4.** Impact of nutrient omission on the agronomic performance of rice, maize, and rice–maize system (3 yr average).

| Treatment | Grain Yield (t ha$^{-1}$) | | System Yield (Rice Equivalent) (t ha$^{-1}$) | SYI for System | Nutrient Response (t ha$^{-1}$) | | | Biomass Yield (t ha$^{-1}$) | | Harvest Index | |
|---|---|---|---|---|---|---|---|---|---|---|---|
| | Rice | Maize | | | Rice | Maize | System | Rice | Maize | Rice | Maize |
| $T_1$ | 6.81 | 9.66 | 16.32 | 0.89 | - | - | - | 13.83 | 19.41 | 0.49 | 0.51 |
| $T_2$ | 2.97 | 3.76 | 6.34 | 0.25 | 3.85 | 5.90 | 9.98 | 7.14 | 7.54 | 0.41 | 0.45 |
| $T_3$ | 3.47 | 4.29 | 7.50 | 0.33 | 3.35 | 5.37 | 8.82 | 7.93 | 8.71 | 0.44 | 0.47 |
| $T_4$ | 5.64 | 7.26 | 12.64 | 0.61 | 1.17 | 2.40 | 3.69 | 11.78 | 14.56 | 0.48 | 0.49 |
| $T_5$ | 5.73 | 7.76 | 13.24 | 0.66 | 1.09 | 1.90 | 3.08 | 12.07 | 15.55 | 0.47 | 0.50 |
| $T_6$ | 6.22 | 8.84 | 14.79 | 0.79 | 0.59 | 0.81 | 1.54 | 12.97 | 17.68 | 0.48 | 0.50 |
| $T_7$ | 6.22 | 8.47 | 14.44 | 0.74 | 0.59 | 1.19 | 1.88 | 13.18 | 16.87 | 0.47 | 0.50 |
| $T_8$ | 2.14 | 2.55 | 4.33 | 0.32 | 2.51 | 3.47 | 5.69 | 6.28 | 5.12 | 0.34 | 0.44 |
| $T_9$ | 4.65 | 6.02 | 10.02 | 0.88 | - | - | - | 11.85 | 12.24 | 0.39 | 0.45 |
| LSD ($p \leq 0.05$) | 0.47 | 0.57 | 0.73 | - | - | - | - | 0.90 | 1.01 | 0.02 | 0.03 |

Refer Table 2 for treatment description; SYI: sustainable yield index.

**Table 5.** (**a**) Effect of nutrient omission on the 3 yr average nitrogen, phosphorous and potassium content (%) in rice and maize crops. (**b**) Effect of nutrient omission on the 3 yr average sulphur (%) and zinc content (mg kg$^{-1}$) in rice and maize crops.

**(a)**

| Treatment | N | | | | P | | | | K | | | |
|---|---|---|---|---|---|---|---|---|---|---|---|---|
| | Rice | | Maize | | Rice | | Maize | | Rice | | Maize | |
| | Grain | Straw | Grain | Stover | Grain | Straw | Grain | Stover | Grain | Straw | Grain | Stover |
| $T_1$ | 1.45 | 0.399 | 1.06 | 1.02 | 0.234 | 0.074 | 0.193 | 0.244 | 0.225 | 1.76 | 0.705 | 2.47 |
| $T_2$ | 1.03 | 0.268 | 0.67 | 0.42 | 0.179 | 0.053 | 0.106 | 0.109 | 0.165 | 1.41 | 0.441 | 1.39 |
| $T_3$ | 1.13 | 0.327 | 0.68 | 0.70 | 0.215 | 0.070 | 0.161 | 0.186 | 0.204 | 1.67 | 0.624 | 2.07 |
| $T_4$ | 1.32 | 0.363 | 0.90 | 0.85 | 0.192 | 0.055 | 0.127 | 0.147 | 0.205 | 1.70 | 0.638 | 2.18 |
| $T_5$ | 1.35 | 0.366 | 0.91 | 0.90 | 0.219 | 0.066 | 0.165 | 0.201 | 0.183 | 1.35 | 0.506 | 1.61 |
| $T_6$ | 1.37 | 0.386 | 0.91 | 0.91 | 0.220 | 0.066 | 0.172 | 0.215 | 0.210 | 1.71 | 0.652 | 2.20 |
| $T_7$ | 1.38 | 0.384 | 0.93 | 0.95 | 0.223 | 0.068 | 0.180 | 0.223 | 0.210 | 1.71 | 0.662 | 2.23 |
| $T_8$ | 1.02 | 0.234 | 0.59 | 0.37 | 0.167 | 0.053 | 0.095 | 0.094 | 0.155 | 1.27 | 0.379 | 1.33 |
| $T_9$ | 1.36 | 0.307 | 0.86 | 0.75 | 0.211 | 0.065 | 0.159 | 0.184 | 0.175 | 1.40 | 0.561 | 1.92 |
| LSD ($p \leq 0.05$) | 0.05 | 0.022 | 0.05 | 0.05 | 0.010 | 0.005 | 0.012 | 0.011 | 0.006 | 0.10 | 0.011 | 0.07 |

**(b)**

| Treatment | S | | | | Zn | | | |
|---|---|---|---|---|---|---|---|---|
| | Rice | | Maize | | Rice | | Maize | |
| | Grain | Straw | Grain | Stover | Grain | Straw | Grain | Stover |
| $T_1$ | 0.110 | 0.118 | 0.132 | 0.193 | 33.9 | 37.7 | 43.8 | 127.0 |
| $T_2$ | 0.092 | 0.101 | 0.082 | 0.101 | 28.7 | 30.7 | 25.6 | 49.8 |
| $T_3$ | 0.103 | 0.109 | 0.120 | 0.177 | 32.0 | 34.6 | 38.1 | 112.2 |
| $T_4$ | 0.107 | 0.113 | 0.122 | 0.184 | 32.7 | 36.4 | 38.8 | 114.9 |
| $T_5$ | 0.107 | 0.114 | 0.128 | 0.186 | 33.2 | 36.4 | 40.4 | 117.1 |
| $T_6$ | 0.098 | 0.101 | 0.093 | 0.136 | 33.0 | 36.2 | 41.5 | 119.6 |
| $T_7$ | 0.107 | 0.113 | 0.130 | 0.188 | 29.1 | 32.3 | 29.4 | 65.6 |
| $T_8$ | 0.083 | 0.097 | 0.079 | 0.096 | 27.4 | 29.1 | 21.9 | 43.8 |
| $T_9$ | 0.097 | 0.112 | 0.126 | 0.173 | 32.1 | 34.8 | 38.1 | 89.0 |
| LSD ($p \leq 0.05$) | 0.005 | 0.006 | 0.006 | 0.008 | 1.9 | 2.7 | 4.1 | 8.7 |

Refer Table 2 for treatment description.

**Table 6.** Effect of nutrient omission on nitrogen (N), phosphorous (P), potassium (K), sulphur (S) (kg ha$^{-1}$) and zinc (g ha$^{-1}$) 3 yr average uptake in grain and straw.

| Treatment | Rice | | | | | Maize | | | | | System | | | | |
|---|---|---|---|---|---|---|---|---|---|---|---|---|---|---|---|
| | N | P | K | S | Zn | N | P | K | S | Zn | N | P | K | S | Zn |
| $T_1$ | 126.8 | 21.1 | 139.1 | 15.8 | 496.2 | 156.5 | 31.5 | 214.4 | 22.1 | 943.8 | 283.4 | 52.7 | 353.4 | 37.9 | 1440.0 |
| $T_2$ | 42.6 | 7.6 | 63.8 | 7.0 | 214.9 | 33.0 | 6.1 | 54.5 | 4.7 | 174.2 | 75.6 | 13.7 | 118.3 | 11.7 | 389.0 |
| $T_3$ | 54.5 | 10.6 | 81.9 | 8.5 | 266.7 | 46.9 | 11.3 | 84.6 | 9.0 | 360.1 | 101.4 | 21.9 | 166.4 | 17.5 | 626.8 |
| $T_4$ | 96.9 | 14.3 | 115.8 | 13.0 | 408.9 | 98.7 | 15.0 | 146.1 | 15.5 | 616.5 | 195.6 | 29.3 | 261.9 | 28.4 | 1025.3 |
| $T_5$ | 100.6 | 16.8 | 95.9 | 13.4 | 422.3 | 107.8 | 21.4 | 118.9 | 17.1 | 685.8 | 208.4 | 38.1 | 214.9 | 30.5 | 1108.1 |
| $T_6$ | 111.2 | 18.2 | 128.6 | 12.8 | 450.4 | 122.8 | 25.4 | 176.9 | 14.2 | 802.5 | 234.0 | 43.6 | 305.6 | 27.0 | 1253.0 |
| $T_7$ | 112.6 | 18.6 | 132.0 | 14.5 | 406.7 | 120.8 | 25.3 | 170.7 | 18.8 | 497.0 | 233.4 | 44.0 | 302.8 | 33.3 | 903.7 |
| $T_8$ | 31.7 | 5.8 | 55.6 | 5.7 | 179.2 | 18.7 | 3.4 | 33.9 | 2.9 | 99.3 | 50.4 | 9.2 | 89.5 | 8.7 | 278.6 |
| $T_9$ | 85.2 | 14.4 | 107.9 | 12.5 | 399.8 | 74.3 | 14.4 | 113.7 | 12.6 | 469.5 | 159.5 | 28.8 | 221.6 | 25.1 | 869.3 |
| LSD ($p \leq 0.05$) | 8.8 | 1.4 | 10.0 | 1.0 | 43.6 | 7.6 | 1.8 | 11.5 | 1.4 | 55.3 | 10.4 | 2.4 | 15.5 | 1.8 | 69.1 |

Refer Table 2 for treatment description.

**Table 7.** Agronomic Efficiency (AE) *, Physiological Efficiency (PE) ** and Apparent Recovery (AR) of applied nutrients by hybrid crops.

| Nutrients | Agronomic Efficiency (kg kg$^{-1}$) | | | | | | Physiological Efficiency (kg kg$^{-1}$) | | | | | | Apparent Recovery (%) | | | | | |
|---|---|---|---|---|---|---|---|---|---|---|---|---|---|---|---|---|---|---|
| | Rice | | | Maize | | | Rice | | | Maize | | | Rice | | | Maize | | |
| | Yr 1 | Yr 2 | Yr 3 | Yr 1 | Yr 2 | Yr 3 | Yr 1 | Yr 2 | Yr 3 | Yr 1 | Yr 2 | Yr 3 | Yr 1 | Yr 2 | Yr 3 | Yr 1 | Yr 2 | Yr 3 |
| N | 16.1 | 19.0 | 22.3 | 20.3 | 28.4 | 32.5 | 49.0 | 45.6 | 45.1 | 50.6 | 52.7 | 52.0 | 32.9 | 41.6 | 49.5 | 40.2 | 54.0 | 62.5 |
| P | 8.7 | 13.1 | 28.3 | 9.0 | 19.3 | 27.9 | 139.8 | 153.4 | 196.8 | 112.2 | 166.8 | 177.7 | 6.2 | 8.6 | 14.4 | 8.1 | 11.6 | 15.7 |
| K | 6.2 | 14.8 | 19.8 | 4.6 | 11.0 | 15.7 | 15.2 | 26.4 | 30.5 | 16.3 | 20.9 | 25.2 | 40.6 | 56.0 | 65.1 | 28.3 | 52.7 | 62.2 |
| S | 8.6 | 16.4 | 34.0 | 19.0 | 31.2 | 48.8 | 148.5 | 173.6 | 241.1 | 90.8 | 117.5 | 151.3 | 5.8 | 9.5 | 14.1 | 20.9 | 26.5 | 32.3 |
| Zn | 95.2 | 146.5 | 351.0 | 178.2 | 466.4 | 704.1 | 5898.5 | 5980.8 | 7168.6 | 1700.6 | 3091.8 | 3683.0 | 1.6 | 2.5 | 4.9 | 10.5 | 15.1 | 19.1 |

* AE: kg grain increase per kg nutrient applied; ** PE: kg grain increase per kg increase in total nutrient accumulation. Refer Table 2 for treatment description.

In general, the physiological efficiency (kg kg$^{-1}$) in the hybrid rice and maize was lower in the first year and increased in the second and third years (except for the N in rice) (Table 7).

The apparent recovery (AR) of N by the hybrid maize and K by the hybrid rice was higher than for the other nutrients during all three years. Additionally, the AR was higher in the 2nd and 3rd years than in the 1st year (Table 7). The 3 yr average AR (%) of nitrogen, phosphorous, potassium, sulphur and zinc by the hybrid rice was 41.3, 9.7, 53.9, 9.8 and 3.0, respectively. The AR (%) for the hybrid maize was more than for the hybrid rice, with the AR (%) of nitrogen, phosphorous, potassium, sulphur and zinc for the hybrid maize being 52.2, 11.8, 47.7, 26.6 and 14.9, respectively (data not depicted in table).

The reciprocal internal use efficiency (RIUE) of nitrogen, phosphorous, potassium, sulphur S (kg t$^{-1}$ grain) and zinc (g t$^{-1}$ grain) for the hybrid rice was, in general, higher in the 2nd and 3rd year compared to the 1st year. The average reciprocal internal use efficiency of nitrogen, phosphorous, potassium, sulphur (kg t$^{-1}$ grain) and zinc (g t$^{-1}$ grain) for the hybrid rice was 18.6, 3.1, 20.4, 2.3 and 72.9, respectively, and 15.9, 3.2, 21.8, 2.3 and 96.0, respectively, for the hybrid maize crop. The average reciprocal internal use efficiency of nitrogen, phosphorous, potassium, sulphur (kg t$^{-1}$ grain) and zinc (g t$^{-1}$ grain) for the conventional rice and maize was 18.1, 3.1, 22.9, 2.7 and 84.7 and 13.4, 2.6, 20.5, 2.3 and 85.0, respectively (Table 8).

**Table 8.** Reciprocal Internal Use Efficiency (RIUE) * of nitrogen, phosphorous, potassium, sulphur (kg t$^{-1}$ grain) and zinc (g t$^{-1}$ grain) for rice and maize crops.

| Nutrient | Rice | | | | | | | | Maize | | | | | | | |
| --- | --- | --- | --- | --- | --- | --- | --- | --- | --- | --- | --- | --- | --- | --- | --- | --- |
| | Hybrid | | | | Conventional | | | | Hybrid | | | | Conventional | | | |
| | Yr 1 | Yr 2 | Yr 3 | Av | Yr 1 | Yr 2 | Yr 3 | Av | Yr 1 | Yr 2 | Yr 3 | Av | Yr 1 | Yr 2 | Yr 3 | Av |
| N | 18.5 | 18.7 | 18.7 | **18.6** | 18.6 | 18.2 | 17.4 | **18.1** | 15.6 | 16.0 | 16.2 | **15.9** | 12.3 | 13.7 | 14.1 | **13.4** |
| P | 3.1 | 3.1 | 3.1 | **3.1** | 3.2 | 3.1 | 2.9 | **3.1** | 3.2 | 3.2 | 3.3 | **3.2** | 2.4 | 2.6 | 2.7 | **2.6** |
| K | 19.9 | 20.3 | 21.1 | **20.4** | 24.7 | 23.6 | 20.4 | **22.9** | 21.3 | 22.3 | 21.8 | **21.8** | 19.1 | 21.2 | 21.1 | **20.5** |
| S | 2.2 | 2.3 | 2.4 | **2.3** | 2.5 | 2.8 | 2.7 | **2.7** | 2.2 | 2.3 | 2.3 | **2.3** | 2.0 | 2.5 | 2.4 | **2.3** |
| Zn | 72.8 | 71.5 | 74.3 | **72.9** | 83.3 | 90.6 | 80.1 | **84.7** | 93.7 | 97.7 | 96.7 | **96.0** | 79.0 | 89.0 | 87.1 | **85.0** |

* Hybrid rice (Arize-6444), conventional rice (Rajshree), hybrid maize (DKC-9081), and conventional maize (Laxmi). Refer Table 2 for treatment description.

### 3.4. Contribution of Available Soil Nutrient toward Hybrid Rice and Maize Yield

The % contribution from the soil available nutrients to the total uptake of major nutrients (N, P and K) by the rice and maize is presented in Figure 2. In general, the % contribution from the soil was higher for the hybrid than for the conventional crops. The contribution of available nitrogen, phosphorous and potassium from the soil towards total removal by the hybrid rice and maize was 20.0, 44.7, and 74.0% and 15.5, 35.7, and 63.2%, respectively. The uptake by the conventional rice and maize was 14.9, 34.0, 64.5, 8.8, 19.9, and 39.3%, respectively. The average N, P and K contributions from the soil towards total removal by the hybrid rice–maize cropping system were 35.5, 80.4 and 137.2%, respectively, and by the conventional rice–maize system were 23.7, 53.9 and 103.8%, respectively.

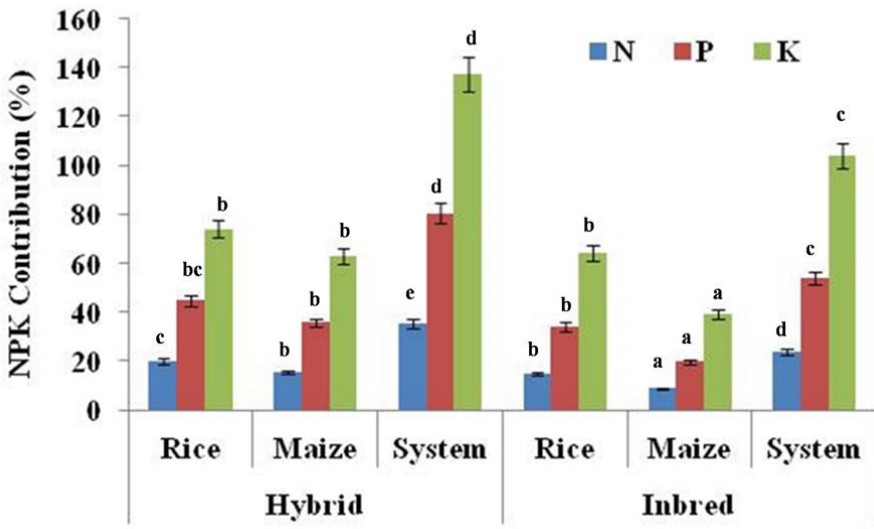

**Figure 2.** Percent contribution of nutrients from available soil nutrients towards total accumulation by rice and maize crops. Similar letters for treatments within each nutrient group are not significantly different at *p* = 0.05 by LSD.

### 3.5. Economics of Nutrient Management

The economic loss for the hybrid rice and maize crops due to nutrient omission is depicted in Figure 3. In general, the grain yield loss in both the hybrid rice and maize crops due to nitrogen omission was highest followed by phosphorous, potassium, zinc and sulphur. The loss was more pronounced during the second and third years than the first year. Additionally, the economic loss due to nutrient omission was more in the hybrid maize than the hybrid rice (Figure 3). The 3 yr average hybrid rice grain yield loss due to the omission of nitrogen, phosphorous, potassium, sulphur and zinc was equivalent to economic losses of USD 782, USD 272, USD 252, USD 137 and USD 138 ha$^{-1}$, respectively. While in the hybrid maize, it was USD 1180, USD 540, USD 428, USD 204 and USD 276 ha$^{-1}$, respectively. The 3 yr average return on investment (ROI) for nitrogen, phosphorous, potassium, sulphur and zinc (i.e., a USD value of grain yield increase per USD

invested in nutrient fertilizer) for the hybrid rice was 21.2, 7.1, 6.7, 4.1 and 0.3, respectively. For the hybrid maize, the ROIs were 28.8, 7.6, 4.9, 6.5 and 0.7, respectively (Figure 4).

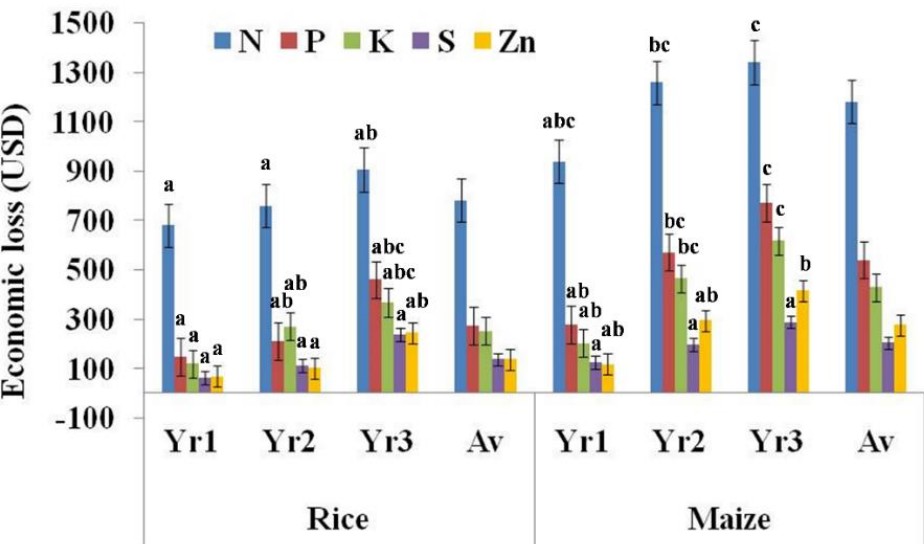

**Figure 3.** Economic loss due to nutrient omission. Similar letters for treatments within each nutrient group are not significantly different at *p* = 0.05 by LSD.

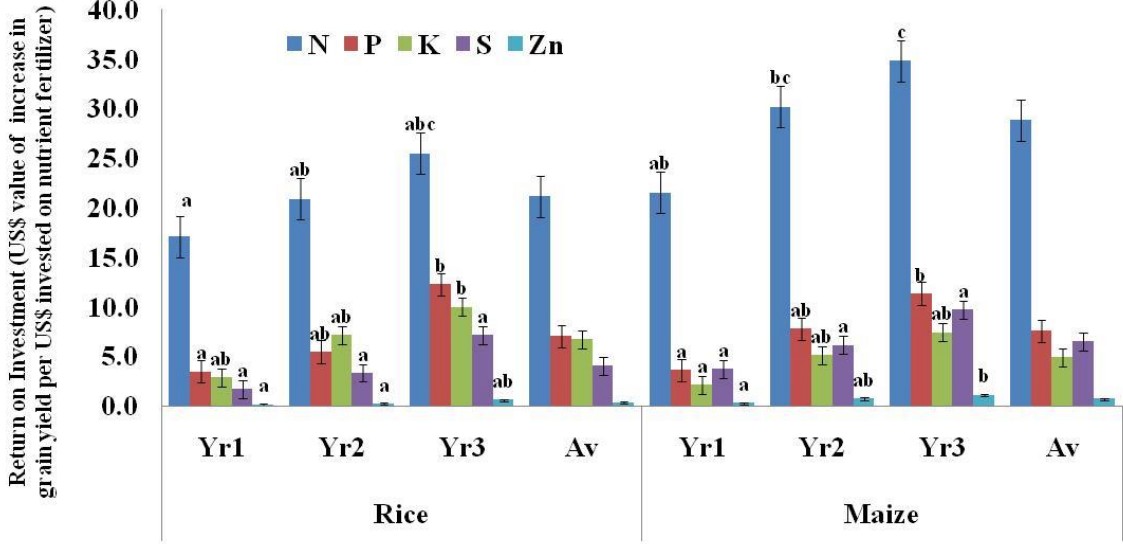

**Figure 4.** Return on investment due to nutrient omission. Similar letters for treatments within each nutrient group are not significantly different at *p* = 0.05 by LSD.

*3.6. Nutrient Status in Post-Harvest Soil*

The status of the soil fertility (0 to 15 cm) collected after the harvest of the third year maize (completion of study) is depicted in Table 9. The soil nutrient depletion over time was significant for N, P, K and Zn in the nutrient-omitted and unfertilized plots compared with the fully fertilized plot. The available nitrogen, phosphorous, potassium (kg ha$^{-1}$), sulphur and zinc (mg kg$^{-1}$) varied from 201.2 to 221.6, 13.4 to 23.5, 76.3 to 93.4, 45.4 to 54.7 and 0.54 to 0.83, respectively, following the different nutrient management practices. There was a decrease over time in the availability of nutrients when comparing the initial soil status to the respective nutrient omission plots. The greatest decrease (18.9 kg ha$^{-1}$) was observed in the N-omitted plot and decreases of 6.1 (kg ha$^{-1}$), 6.8 (kg ha$^{-1}$), 5.4 (mg kg$^{-1}$) and 0.15 (mg kg$^{-1}$) occurred in the phosphorous, potassium, sulphur and zinc-omitted plots, respectively. A build-up in the available nutrients, viz., N, P, K, S and Zn, was 8.8, 6.5, 7.2 kg ha$^{-1}$, 1.9 and 0.11 mg kg$^{-1}$ recorded over the initial status of the respective nutrient.

**Table 9.** Impact of nutrient management on the status of post-harvest soil (0 to 15 cm) after 3 yrs of nutrient omission.

| Treatment | pH (1:2, Soil:Water) | EC (1:2, Soil:Water) dSm$^{-1}$ | OC (%) | N (kg ha$^{-1}$) | P (kg ha$^{-1}$) | K (kg ha$^{-1}$) | S (mg kg$^{-1}$) | Zn (mg kg$^{-1}$) |
|---|---|---|---|---|---|---|---|---|
| $T_1$ | 8.11 | 0.445 | 0.37 | 221.6 | 23.5 | 93.4 | 54.7 | 0.83 |
| $T_2$ | 8.24 | 0.463 | 0.30 | 201.2 | 13.4 | 76.3 | 45.4 | 0.54 |
| $T_3$ | 8.22 | 0.460 | 0.30 | 193.9 | 20.7 | 89.2 | 53.0 | 0.78 |
| $T_4$ | 8.24 | 0.457 | 0.29 | 216.4 | 10.9 | 91.0 | 52.7 | 0.77 |
| $T_5$ | 8.19 | 0.459 | 0.32 | 216.8 | 21.0 | 79.5 | 53.9 | 0.78 |
| $T_6$ | 8.21 | 0.448 | 0.34 | 217.4 | 21.5 | 90.1 | 47.4 | 0.77 |
| $T_7$ | 8.22 | 0.449 | 0.33 | 216.2 | 21.7 | 90.0 | 54.0 | 0.57 |
| $T_8$ | 8.27 | 0.468 | 0.30 | 203.3 | 14.7 | 78.5 | 46.5 | 0.59 |
| $T_9$ | 8.18 | 0.451 | 0.34 | 218.1 | 21.9 | 88.2 | 54.6 | 0.83 |
| LSD ($p \leq 0.05$) | NS | NS | 0.04 | 17.6 | 1.7 | 8.5 | 4.9 | 0.09 |

Refer Table 2 for treatment description.

## 4. Discussion

### 4.1. Yield and Associated Parameters

The grain and straw/stover yield of the hybrid rice and maize was found to decrease in all the nutrient-omitted plots over the optimum fertilized plot (Table 4). The decrease in grain yield was higher during the third year than the second and first years, which might be due to the regular omission of the respective nutrients. In a study with nutrient omissions for rice, the most limiting nutrient was nitrogen followed by phosphorous and potassium [12,35]. Salam et al. [35] observed a 20–26, 9–13 and 5–9% rice yield penalty due to the omissions of N, P and K, respectively. The decline in productivity due to nutrient omission also depends upon the soil environment during the growing period of a crop [12].

In nutrient omission on-farm trials conducted across the world, a significant yield penalty due to nutrient omissions was shown [44,45], and all these studies explained the role of nutrients in the growth and yield of the maize plant. In another study, Ray et al. [46] also measured the impact of nitrogen, phosphorous, and potassium omissions along with an absolute control on the grain yield as well as biomass. They reported that a nitrogen omission caused a greater reduction in the grain yield, followed by K and P omissions, respectively. The nitrogen, phosphorous and potassium limited grain yields were estimated at 5.29, 7.85 and 6.89 t ha$^{-1}$, respectively, compared to 8.34 t ha$^{-1}$ at an ample NPK application. The authors mentioned that the prevailing P × K, N × K and N × P interactions in soil could be a reason for this in addition to the regular role of nutrients for crop growth. Furthermore, the authors reported that, when compared with a 100% recommended dose of fertilizer (200:60:60::N:P$_2$O$_5$:K$_2$O), the grain yield reductions with nutrient omissions were about 44% for nitrogen, 17% for phosphorous, and 27% for potassium omissions. In another study conducted in the central region of Togo, a decrease in the maize yield was observed due to a nitrogen, phosphorous, potassium, sulphur and zinc omission [44]. The three-year results of the NPK response trial revealed that a balanced application of nutrients resulted in a higher crop productivity. Moreover, omitting N caused a 56.2% reduction of the rice grain yield, while the omission of P and K resulted in reductions of 25.4 and 6.0%, respectively [47].

The sustainability yield index was highest for hybrid and conventional crops in the optimum-fertilized (NPKSZn) plots (Table 4). Nutrient omission reduced the yield sustainability of the system. The omission of N reduced the sustainability the most, followed by phosphorous, potassium, zinc, and sulphur. Among the nutrient omission plots, the SYI was affected the least following omission because of the high available sulphur in the initial soil (Table 1). The SYI for the hybrid crops was lower than the conventional crops in the unfertilized plot. The findings showed that the application of balanced nutrition resulted in a greater yield sustainability for the hybrid and conventional rice–maize cropping systems. The treatments without an adequate supply of essential plant nutrients through fertilizers resulted in a lower sustainable yield index than the treatments that received the recommended doses of mineral fertilizers [48,49].

Our study showed that under ample fertilization, the hybrid rice and maize crop yielded a 2.2 and 5.6 t ha$^{-1}$ higher grain than the conventional crops, respectively, which is attributed to a higher accumulation of dry matter and a higher harvest index, as suggested in the previous studies [50,51]. Moreover, hybrid rice normally has a yield advantage of 20–30 per cent over conventional cultivars [52] and an advantage of 1 to 1.5 t ha$^{-1}$ of hybrid rice varieties over semi-dwarf, conventional, high-yielding varieties in the farmers' fields in China and other countries [53], but these findings showed more yield gain than the reported yields. Islam et al. [51] conducted an experiment at the experimental farm of the International Rice Research Institute (IRRI), Los Baños, Philippines, and reported a higher grain yield of hybrid rice compared to conventional rice; thus, the selection of suitable hybrids coupled with proper nutrient management may help to enhance the yield of crops. In line with this, the present study also highlights that the yields of the hybrids decreased significantly without receiving any nutrients, and the decrease was even greater than that of the conventional crops; therefore, the use of hybrids can only be efficient with a balanced nutrient management depicting the synergistic effect of hybrids and nutrient management.

The harvest index (HI) is a useful parameter to assess the nutrient translocation efficiency. The balanced application of nutrients in our study favoured more grain yield over straw yield and, thus, increased the proportion of economic yield which resulted in an improved HI (Table 4). Hybrids have been reported to maintain higher photosynthesis, a greater translocation of nutrients, and HI and stress tolerance at the grain filling stage [45,54–57]; thus, resulting in an improvement in the nutrient harvest index and increased internal efficiency (IE) as compared with conventional crops.

The agronomic efficiency (AE) of different nutrients depicts the kg grain yield increase per kg of the nutrient applied. The AE of the nutrients was higher for the maize than the rice, which might be attributed to the higher yield response of maize due to the applied fertilizer nutrients [12,58,59].

The AE was more in the third year than the second year and first year in both crops, which might be due to an increase in the differences in yield among the ample fertilized plots and respective nutrient-omitted plots, and also due to a variability in the release of inorganic nitrogen from the soil [60]. The physiological nutrient use efficiency (PE) for the grain or total biomass production per unit nutrient removal by the plants is a key parameter for evaluating the nutrient use efficiency in rice and maize. Similar to agronomic efficiency, the physiological efficiency of the nutrients was greater for the maize crop than that for the rice and it was also greater in the third year than the second and first years in both crops (Table 7). The improvement in the nutrient use efficiencies might have been due to better nutrient management [61]. Salam et al. [35] conducted field experiments in rice-based cropping systems at Gazipur, Bangladesh under nutrient omission and reported more agronomic efficiency and physiological efficiency of nutrients during the second year than the first year. The reciprocal internal use efficiency was slightly more in hybrid than conventional crops, which might be due to a higher harvest index of the hybrid crops. The yield potential of hybrid crops is higher than that of conventional crops; thus, for optimum yield production, hybrids require more nutrition than conventional crops.

Dobermann [62] reported that the agronomic efficiency of nitrogen in cereals ranged from 10–30 kg per kg and can reach >30 kg per kg only in the best-managed systems with low levels of fertilizer N or with a low soil N supply. The internal efficiency (IE) indicates the ability of plants to transform nutrients taken up from the soil and fertilizer into the economic yield [63]. A low internal efficiency suggests poor internal nutrient conversion due to stress (i.e., nutrient deficiencies, drought, heat, mineral toxicities, and disease). The trend of the IE of nutrients (i.e., P > K > N) was similar to the earlier finding by Liu et al. [64]. Pathak et al. [65] reported the average internal efficiency of nitrogen ($IE_N$), internal efficiency of phosphorous ($IE_P$), and internal efficiency of potassium ($IE_K$) in cereal-based systems as 20.9–65.9, 131.9–402.8 and 18.3–96.3 kg kg$^{-1}$, respectively. In Indian soils, the average $RE_N$, $RE_P$ and $RE_K$ were 36.3–79.4, 12.4–46.7 and 4.5–71.8%, respectively, from 1970 to 1998 [65]. The applied fertilizer nutrients not taken up by the crops are vulnerable

to losses through leaching, erosion, and denitrification or volatilization (for nitrogen only), and all of these influence the recovery efficiency [64]. In the farmers' fields, the average N recovery efficiency ranged from about 20 to 30% under rainfed conditions and 30 to 40% under irrigated conditions [61,66,67]. Nitrogen recovery rarely exceeds 50% in crops grown by farmers and is generally much lower [68]; however, in the present study, the values of $RE_N$ were higher than those reported by the previous authors. Recently, Ladha et al. [69] has also reported that the single-year fertilizer nitrogen recovery efficiencies averaged 65% for maize. The low recoveries could be related to denitrification, volatilization, and the leaching losses of nitrogen from the soil [70]; however, information regarding the P and K recovery in maize is meagre.

### 4.2. Nutrient Uptake and Recovery Efficiency

The nutrient concentration in grain and straw/stover, and uptake by the rice and maize decreased in the nutrient-omission and unfertilized control plots over the optimum fertilized plots (Table 5a,b). The highest nutrient concentration and uptake in the ample fertilized plot was attributed to the higher available nutrients and possibly more proliferation of the root system with balanced fertilization, leading to the higher absorption of nutrients from the soil. The lowest nutrient uptake was recorded in the N omission followed by P, K, Zn and S in the rice–maize cropping system; thus, the rate of dry matter production or grain yield seems to control nutrient uptake. Setiyono et al. [45] observed the lowest nutrient content in maize mainly in nutrient-omitted plots. The authors reported higher nutrient accumulation as compared to our study. On average, the nutrient removal through above-ground plant dry matter was 232 kg N ha$^{-1}$, 35 kg P ha$^{-1}$ and 269 kg K ha$^{-1}$ [45]. These values were higher than our findings, which might be due to the higher average yield (12 t ha$^{-1}$) of the crop grown in their study. The total nutrient uptake was higher in hybrid maize as compared to hybrid rice and that might be due to a higher biomass accumulation in maize. The apparent recovery of nutrients was more in the third year than the second and first years in both rice and maize and this might be due to an increase in the difference of the nutrient uptake by the total biomass in respective nutrient-omitted plots in the third year than the first and second year. The AR of the nutrients by the hybrid maize was more than that of the hybrid rice and that might have been due to more biomass of the maize than the rice, thus accumulating more nutrients than the rice (Table 7).

### 4.3. Nutrient Contribution towards Total Uptake by Crops

Among different nutrients, the maximum % contribution of potassium from the available soil nutrients towards total uptake by both the hybrid and conventional crops was observed followed by phosphorous and nitrogen. The nitrogen, phosphorous and potassium contributions from the available nutrients towards total uptake by the hybrid rice–maize system was more than the conventional rice–maize system (Figure 2). The contributions of the nutrients from soil is governed by different factors, viz., the soil environment, type of crop, rooting pattern, etc. The nutrient contribution from the soil was more in the hybrid crops than the conventional crops and that might have been due to better root growth, leaf area and nutrient translocation in the hybrid crops. Prasad et al. [42] studied the nutrients contribution towards the total crop uptake for development of target yield equation and recorded the maximum contribution of potassium followed by phosphorous and nitrogen in rice, while in maize, it was the maximum for phosphorous followed by potassium and nitrogen. This variation in the soil contribution towards a total uptake might have been due to variations in the crop type and soil environment.

### 4.4. Economic Loss Due to Nutrient Omission

The economic loss in both the hybrids due to nutrient omission was lower during the first year and it was found to increase during the subsequent years (Figure 4). More economic loss during the second and third years was due to a higher yield loss under the regular omission of a nutrient. The economic loss due to the omission of nitrogen was

higher than the other nutrients and this shows that nitrogen plays a more crucial role for crop yields than do the other nutrients [9,12]. The economic loss in the hybrid maize was more as compared to the hybrid rice, which might have been due to a higher economic yield and higher loss due to the nutrient omission in the hybrid maize rather than the hybrid rice. Likewise, the return on investment (ROI) (i.e., USD per USD invested in nutrient fertilizer) was more for the N-omitted plot followed by phosphorous, potassium, sulphur and zinc. In on-farm experiments at different locations in India, the blanket recommendation of fertilizer nutrients causes more economic loss than a balanced management of nutrients [71].

*4.5. Impact on Nutrient Status in Postharvest Soil*

Balanced fertilization in hybrid and conventional crops improved the soil fertility status over nutrient-omitted and unfertilized plots (Table 9). The extent of decline in the nutrient availability in the unfertilized control plot was more in the hybrid grown plots than in the conventionally grown plots, indicating that the hybrid crops showed a higher nutrient uptake from the soil as compared to the conventional crop. In general, the nutrient availability in the postharvest soil decreased more in the respective nutrient-omitted plots than the unfertilized check plot, and this might be due to more yields in the nutrient-omission plot than the unfertilized plot; thus, mining more nutrients from the nutrient-omission plot than the unfertilized plot. The inherent nutrient supplying capacity of a given soil is specific, but that capacity is crop-specific due to variability in crops' biological demands and abilities for nutrient uptake [35,72]. The root morphology and nutrient solubilizing capacity help the different crop species to acclimatize under the stress conditions of nutrient loss and the absorption of soil nutrients [12,35].

## 5. Conclusions

Nutrient availability and crop uptake depends on the nutrient supplying capacity of soils, which was assessed under the present study through an omission plot technique. The findings indicated that the crop yield declined due to nutrient omissions and that the most limiting nutrient was nitrogen followed by phosphorous, potassium, zinc and sulphur. The continuous ignorance of a particular nutrient from a fertilizer schedule will intensify the decline in the yield. The hybrid rice and maize produced a 32 and 38% higher yield than the respective conventional crops with target yield-based fertilization. The sustainable yield index indicated that the hybrid and conventional rice–maize cropping system was more sustainable with balanced fertilization than the respective nutrient omission. The percentage of nutrient contributions from calcareous soils towards total removal by the hybrid crops was higher than the conventional crops, which indicated that the hybrid crops mined more nutrients than the conventional crops from the soil. The reciprocal internal use efficiency was higher in the hybrid than in the conventional crops. The maximum economic loss and lowest internal efficiency (IE) due to a N omission rather than the other nutrients showed that the N plays a crucial role in the crop yield; therefore, in the present case, the approach on nutrient omission for an assessment of the inherent supplying capacity of the nutrients from calcareous soils could be used for improving nutrient prescription, through the synchronization of the nutrient supplying capacity of soil and the yield potential of a crop to enhance the yield, nutrient use efficiency and farm income.

**Author Contributions:** Conceptualization, design and methodology of the experiment, S.P.S., S.D. and K.M.; software used for analysis, S.S.P.; validation and data curation, S.P.S., S.J. and S.K.C.; writing—original draft preparation, S.P.S. and S.D.; writing—review and editing, P.S., P.S.B., K.M.S. and K.K.; supervision, M.C.M. and P.S.B. All authors have read and agreed to the published version of the manuscript.

**Funding:** International Plant Nutrition Institute (IPNI)-South Asia Program (IPNI-2013-IND-523) funded the conduct of the experiment.

**Institutional Review Board Statement:** Not applicable.

**Informed Consent Statement:** Not applicable.

**Data Availability Statement:** Not applicable.

**Acknowledgments:** The authors thankfully acknowledged the technical and administrative support of Rajendra Prasad Central Agricultural University, Bihar, India in the smooth running of the research work. The authors would also like to thank A. M. Johnston, Ex-Vice President, IPNI, Asia, Africa and Middle East Group for his kind interactions and guidance during the execution of the field trial and to Steve Phillips, Principal Scientist, African Plant Nutrition Institute, Lot 660, Hay Moulay Rachid, Benguerir, Morocco for reviewing the manuscript.

**Conflicts of Interest:** The authors declare no conflict of interest.

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
