# Peer review of "Indigenous Nutrient Supplying Capacity of Young Alluvial Calcareous Soils Favours the Sustainable Productivity of Hybrid Rice and Maize Crops"

_sustainability, doi:10.3390/su141811585_

Round 1

Reviewer 1 Report

General comments

The topic of research is relevant for the Sustainability. Authors have aimed for a systematic research for sustainable productivity of hybrid rice and maize. After going through the manuscript, there were some minor corrections which need to be addressed before acceptance.

Specific comments

Ø  The yield gap and research hypothesis needs to be discussed in the introduction part

Ø  Merged words need to be rectified

Ø  “Kharif” and “rabi” needs to be italicized

Ø  In results, yield increment may be expressed in percentage

Ø  Agronomic and physiological efficiencies need to be discussed with proper reasoning

Ø  Economic loss may be supported with other research findings

Ø  Plagiarism needs to be checked before final submission

Ø  Check the citation style, journal name needs to be italicized

Ø  References needs to be cross checked with the text

Author Response

The reply is added for your kind perusal. 

Reviewer 2 Report

Reviewed manuscript “Indigenous nutrient supplying capacity of young alluvial cal-
careous soils favours sustainable productivity of hybrid rice
and maize crops
is an original and interesting study. Authors comprehensively demonstrated the Indigenous nutrient supplying capacity of young alluvial cal-
careous soils for sustainable productivity
. I would suggest minor revision.

Following are some suggestions for further improvements:

First few lines of the abstract should be about the importance of study.

Lines 53-55: Consider revising the lines.

Journal format need to be followed regarding citations write up in text. I thoroughly checked the manuscript there is no MDPI format.

Introduction and Discussion section needs to further strengthen by latest studies on the subject.

At some places in the text, there are grammatical mistakes that need to be corrected by some native English colleague.

To further strengthen introduction, following latest studies etc. are suggested to cite for the importance of alluvial calcareous soils.

Author Response

Reply is added herewith for your kind perusal

Reviewer 3 Report

Dear Editor,

Subject: Review for Sustainability

General comments

The present study, "Indigenous nutrient supplying capacity of young alluvial calcareous soils favours sustainable productivity of hybrid rice and maize crops", conducted by Shiveshwar et al., assesses the nutrient supplying capacity of a calcareous soil using an omission plot technique. I consider the topic is still relevant, as quantifying and understanding the effect of different Nitrogen, Phosphorus, Potassium, Sulphur and Zinc; unfertilized check and omissions of N, P, K, S and Zn in rice and maize. As the authors reported, the application of chemical fertilizer in combination with wheat residue retention is a good strategy to increase energy use efficiency while decreasing carbon footprint in wheat production.

The manuscript directly falls within the scope of the journal and has important deliveries that will benefit the reader of the journal. The manuscript has many grammatical and spacing mistakes, furthermore, the authors need to improve the introduction, results and discussion sections. The authors also needed to use the abbreviation once they used the full name, and need not repeatedly use the full name. However, after careful reading I have some comments and suggestions that should be taken into account when revising this manuscript:

Comments and Suggestions for Authors:

Line 28. Please replace the hyphen “-” of “kg ha-1” with minus sign “⁻”. Please check the entire manuscript and correct it.

Line 28. Add space “_” between “and 104 kg ha-1” and “in the”  

Line 30. Replace the repeated word “for” with “of”

Line 28-31. The sentence is not clear “The three year average return on investment (ROI) for N, P, K, S and Zn (i.e., US$ value of grain yield increase per US$ invested on nutrient fertilizer) for hybrid rice and maize was 21.2, 7.1, 6.7, 4.1, 0.3; and 28.8, 7.6, 4.9, 6.5, 0.7, respectively”. Please rewrite the sentence.

Line 18-36. The abstract section needs improvement.

Line 18-36. The authors need to add one or two sentences about yield in the abstract section because yield is the main purpose of sustainable agriculture.

Line 45. Do you mean rotation of “Rice-maize” if yes, please add rotation after “Rice-maize”.

Line 46. Please add space “_” between “than” and “750”. There have a lot of space mistakes, please recheck the entire manuscript.

Lines 47-48. Once again, the space mistake, please restructure the sentence or replace it with the suggested one “On the other hand, maize (Zea mays L.) is widely cultivated over 150 million ha across 160 countries”.

Line 51-52. Add space between “maize” and “is”, and “being” and consumed.

Line 44-93. The introduction section is very poor, I strongly suggest improving the introduction section.

Read the latest literature and update your old citation with the latest one https://doi.org/10.1080/03650340.2022.2026931 ; https://doi.org/10.3390/agronomy12040845; doi: 10.3389/fmicb.2022.823963; https://doi.org/10.1007/s42729-022-00969-8

Line 158-168. I suggest that the authors need to insert the equation from the “Insert” and “Equation” in word.

Line 204-205. The authors need to use the abbreviation for “nutrient, nitrogen, phosphorous, potassium, sulphur and zinc” once you have used the full name with abbreviations. Please check throughout the manuscript.

Line 226. Replace the word “percent” with “%”

Line 419. Please use the same units throughout the manuscript, in some cases the authors used “kg/kg” and in some cases used “kg kg-1”. Similarly, in some cases “%” and in some cases “percent”. Must be consistence and use the same style for all the units.

Conclusions.  The conclusion section is poor, it should be concise and informative. The author needs to improve the language of the entire manuscript.

Table 1. The sum of the sand, silt, and clay is 98.91%, what about the missing 1.09%? Second please confirm the SOC content is 0.33% correct. The bulk density of the soil is low means the SOC content should be high and need to recheck.

Author Response

Reply is added for your kind perusal

Reviewer 4 Report

Dear Editors,

 I would like to thank you very much for the invitation as a reviewer for the manuscript Sustainability-1882027 “Indigenous nutrient supplying capacity of young alluvial calcareous soils favours sustainable productivity of hybrid rice and maize crops” The article is interesting and dedicated to the vital problem of sustainable grain production. The authors did a big job during this study, which should be highly appreciated.

The authors have to check the compliance of the manuscript with the rules of the journal. I hope that my remarks will help to improve some points of this article.

Lines 90-93 - the aim of the study, should be more clearly stated.

Line 98 – There is only one parenthesis ( .

Line 103 - It would be better for readers if authors put the Figures and Tables not far from the corresponding text paragraphs here and further in text.

Line 106 – “The net plot size was 18 m2  It is not quite clear – is this one treatment or one replication.

Line 141 – Here and further in text: It seems that authors have to inform about the source country of the equipments.

Line 209 – The significance of “the nutrient response” is not indicated in the Table 4. It is not quite clear how these values could be compared.

Lines 262-290 – In the subsection “Nutrient use efficiency” the authors compare different values (AE, AR …) from Tables 7 and 8. These tables do not indicate the significance of the differences between the indicators.

Lines 497-498 – “Thus, the shortfall of the one nutrient affects the uptake of other nutrient by the crops” - This is a general conclusion? And this information has been known for a long time.

Lines 498-500 – “The nutrients omission underlines the significant reduction in the system yield and the most limiting nutrient was nitrogen followed by phosphorous, potassium, zinc and sulphur.” The nutrients omission underlines the significant reduction in the system yield and the most limiting nutrient was nitrogen followed by phosphorous, potassium, zinc and sulphur.  

  Line 501 – Reference. Journal references must cite digital object identifier (DOI) where available (for instance, source No. 17.

Author Response

Reply is added for your kind perusal. 

Round 2

Reviewer 3 Report

Most of the comments have been incorporated
